# Optical Classification of the Remote Sensing Reflectance and Its Application in Deriving the Specific Phytoplankton Absorption in Optically Complex Lakes

**Kun Xue [1], Ronghua Ma [1,\*], Dian Wang [1,2] and Ming Shen [1,2]**

1   Key Laboratory of Watershed Geographic Sciences, Nanjing Institute of Geography and Limnology, Chinese Academy of Sciences, Nanjing 210008, China; kxue@niglas.ac.cn (K.X.); dwang@niglas.ac.cn (D.W.); mshen@niglas.ac.cn (M.S.)
2   University of Chinese Academy of Sciences, Beijing 100049, China
\*   Correspondence: rhma@niglas.ac.cn

**Abstract:** Optical water types (OWTs) were identified from remote sensing reflectance ($R_{rs}(\lambda)$) values in a field-measured dataset of several large lakes in the lower reaches of the Yangtze and Huai River (LYHR) Basin. Four OWTs were determined from normalized remote sensing reflectance spectra ($NR_{rs}(\lambda)$) using the $k$-means clustering approach, and were identified in the Sentinel 3A OLCI (Ocean Land Color Instrument) image data over lakes in the LYHR Basin. The results showed that 1) Each OWT is associated with different bio-optical properties, such as the concentration of chlorophyll-$a$ (Chl$a$), suspended particulate matter (SPM), proportion of suspended particulate inorganic matter (SPIM), and absorption coefficient of each component. One optical water type showed an obvious characteristic with a high contribution of mineral particles, while one type was mostly determined by a high content of phytoplankton. The other types belonged to the optically mixed water types. 2) Class-specific Chl$a$ inversion algorithms performed better for all water types, except type 4, compared to the overall dataset. In addition, class-specific inversion algorithms for estimating the Chl$a$-specific absorption coefficient of phytoplankton at 443 nm ($a^{*}_{ph}(443)$) were developed based on the relationship between $a^{*}_{ph}(443)$ and Chl$a$ of each OWT. The spatial variations in the class-specific model-derived $a^{*}_{ph}(443)$ values were illustrated for 2 March 2017, and 24 October 2017. 3) The dominant water type and the Shannon index ($H$) were used to characterize the optical variability or similarity of the lakes in the LYHR Basin using cloud-free OLCI images in 2017. A high optical variation was located in the western and southern parts of Lake Taihu, the southern part of Lake Hongze, Lake Chaohu, and several small lakes near the Yangtze River, while the northern part of Lake Hongze had a low optical diversity. This work demonstrates the potential and necessity of optical classification in estimating bio-optical parameters using class-specific inversion algorithms and monitoring of the optical variations in optically complex and dynamic lake waters.

**Keywords:** optical water types; remote sensing reflectance; specific inherent optical properties; Sentinel 3A/OLCI

## 1. Introduction

Inland lakes not only supply fresh water and food, but also influence the regional climate and ecological environment, such as the hydrological cycle and nutrient dynamics [1]. Ocean color remote sensing has been widely used to monitor the temporal and spatial bio-optical dynamics of inland waters using satellite data. However, the remote sensing reflectance ($R_{rs}(\lambda)$) showed large

variability and dynamics in turbid and eutrophic lakes due to the frequency influence of sediment resuspension, river inflow, and presence of algal blooms. The high complexity and dynamics of bio-optical properties added challenges in the remote sensing inversion process. Therefore, a number of local and regional bio-optical inversion algorithms have been developed to estimate the concentrations of suspended particulate matter (SPM) and chlorophyll-a (Chl*a*), and the inherent optical properties (IOPs) in optically complex lakes [2–5]. However, it is difficult to define the applicability range of these specific or local bio-optical models [6]. The optical variations in the water components, e.g., non-algal particulates (NAP) and phytoplankton, would affect the performance of the local empirical algorithms or semianalytical algorithms in these lakes [7]. In addition, Dall'Olmo and Gitelson (2006) [8] suggested that Chl*a*-specific absorption coefficients, affected by package effects and pigment accumulation, are also an important factor. It is not feasible to develop a universe algorithm to derive bio-optical parameters in optically complex waters with multiple water types, such as phytoplankton-dominated waters and colored dissolved organic matter (CDOM)-dominated twaters [9,10].

An effective way to improve the remote sensing inversion algorithms in optically complex waters is optical classification, which aims to realize the clustering of waters with similar optical properties and development of a suitable algorithm for each optical water type (OWT) [11]. Oceanic waters were first distinguished into two basic water types: Case I and Case II, based on the covariation between phytoplankton and the other water constitutes, e.g., NAP and CDOM. The following studies have treated the partition of oceanic, coastal, and lake waters into different optical classes based on field-measured or satellite remote sensing reflectance [10,12], inherent optical properties [13], or specific absorption coefficients [14]. Clustering techniques, such as hierarchical clustering [15], *k*-means [16,17], fuzzy *c*-means [12,18], ISODATA (Iterative Self-Organizing Data Analysis Technique) [19], and self-organizing maps [20], were used to partition waters into different groups based on the magnitudes and spectrum characteristics of $R_{rs}(\lambda)$. Optical classification frameworks or schemes of global oceanic [21], coastal [19], and inland waters [16,22] were established using large datasets collected globally. In addition, the optical classification method was also often used to build class-specific bio-optical models in regional coastal [11,23] and inland waters [9,15,24].

Previous studies have demonstrated that optical classification improved the inversion of bio-optical parameters [11], the identification of specific phytoplankton [25] or phytoplankton groups [20,26] and the characterization of the uncertainties associated with ocean color products [27,28]. For example, improvements in estimating the SPM and Chl*a*, especially for Chl*a* below 30 mg/m$^3$, using an optical classification method, were observed in turbid and eutrophic lakes (e.g., Lake Taihu and Lake Chaohu) [9,17]. In addition, a red band-based water classification approach was also provided to improve the performance of the Chl*a* inversion algorithms with a general improvement in mean absolute percentage error (MAPE) by 8.4% for optically complex estuaries [29,30].

The large range and variability in bio-optical parameters (e.g., the concentration of particles, absorption coefficients of water constitutes) were examined in the lower reaches of the Yangtze and Huai River (LYHR) Basin [3,31]. However, whether the optical classification is applicable to lakes in the LYHR basin requires further research. In this study, using both field-measured and OLCI data, optical classification using *k*-means method was tested to demonstrate whether water optical classification is beneficial for improving class-specific bio-optical inversion models in optically complex lakes in the LYHR basin. This study aims to 1) identify the optical water types of the lakes in the LYHR Basin using field-measured and OLCI-derived $R_{rs}(\lambda)$ data; 2) characterize the bio-optical properties and IOP variations in each OWT; and 3) develop class-specific models to improve the estimation of the Chl*a* content and Chl*a*-specific phytoplankton absorption at 443 nm ($a_{ph}^*(443)$).

## 2. Data and Methods

### 2.1. Field-Measured Datasets

We assembled datasets, including hyperspectral $R_{rs}(\lambda)$ data and concentrations of optical active constitutes (OACs), and absorption coefficients of each component (phytoplankton, NAP, and CDOM) from several large lakes in the LYHR Basin. The dataset contained 535 water samples, collected from 2011 to 2017, from Lake Taihu, Lake Chaohu, Lake Hongze, and Lake Shijiu in the LYHR Basin (Figure 1). The lakes in the LYHR Basin are mostly eutrophic and turbid due to the frequent occurrence of algal blooms and sediment resuspension [2,32].

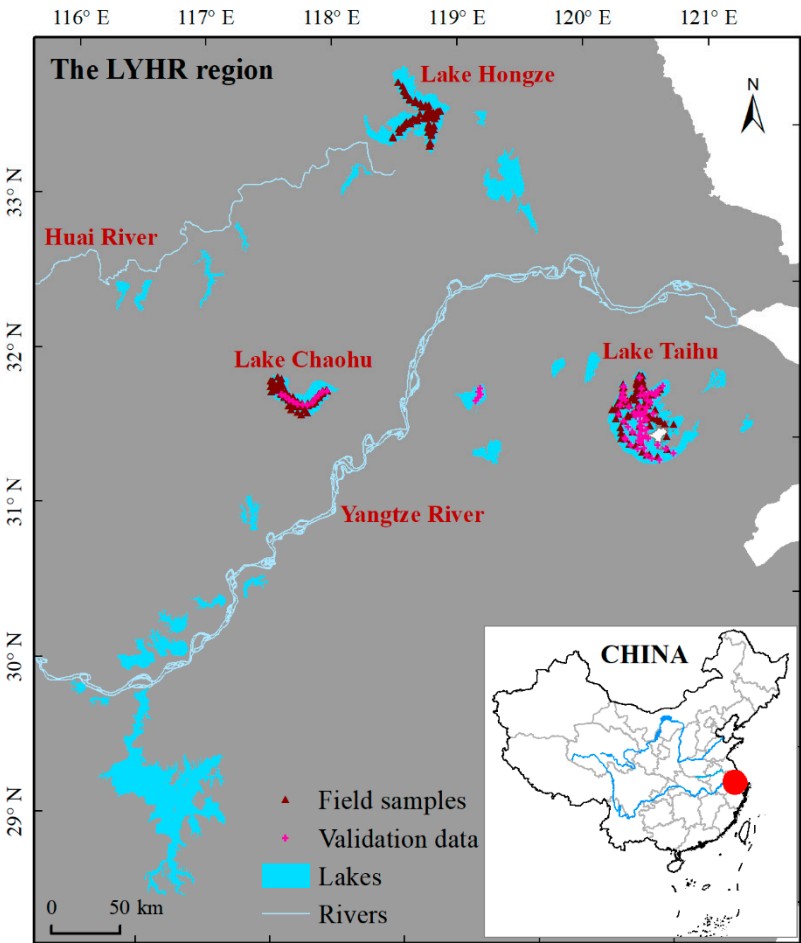

**Figure 1.** Location of the lakes in the lower reaches of the Yangtze and Huai River (LYHR) Basin. The field samples of Lake Chaohu, Lake Taihu, and Lake Hongze were collected from 2011 to 2017. The validation data were match-up pairs of field data and Ocean Land Color Instrument (OLCI)-derived data.

The remote sensing reflectance ($R_{rs}$, sr$^{-1}$), ranging from 350 to 1050 nm, was measured using the above-water method (FieldSpec Pro Dual VNIR, Analytical Spectral Devices, Inc.) [33]. According to the study of Mobley [34] and measurement conditions (viewing direction of 40° from the nadir and 135° from the Sun), the value of the reflectance ratio $\varrho = 0.028$ was used to derive $R_{rs}$ from the measured total water-leaving radiance ($L_{sw}$), radiance of the gray panel ($L_p$), and sky radiance ($L_{sky}$). $R_{rs}$ of validation data was derived using $\varrho$ from the look up table of Mobley (2015) [35]. The water samples were collected near the water surface (<0.3 m) and were stored in the dark at 4 °C before laboratory analysis. According to NASA-recommended protocols, the concentration of Chl*a* was measured spectrophotometrically using a Shimadzu UV-2600 spectrophotometer [36,37]. The concentrations

of SPM were determined gravimetrically in the laboratory, and were further differentiated into the suspended particulate inorganic matter (SPIM) and suspended particulate organic matter (SPOM) by burning the organic matter from the filters [38].

The spectral absorption coefficients of the particulates involved phytoplankton ($a_{ph}(\lambda)$), NAP (also referred to as the detritus) ($a_d(\lambda)$) were determined using the quantitative filter technique [39]. The spectral absorption coefficients of the CDOM ($a_g(\lambda)$) were determined using a Shimadzu UV-2600 spectrophotometer with Milli-Q water as reference. The absorption coefficient of pure water ($a_w(\lambda)$) was obtained from Pope and Fry [40]. The specific phytoplankton absorption coefficient ($a^*_{ph}(\lambda)$) was the ratio of $a_{ph}(\lambda)$ and Chl*a*, and the specific NAP absorption coefficient ($a^*_d(\lambda)$) was the ratio of $a_d(\lambda)$ and SPM [41,42]. Note that the definitions of $a^*_{ph}(\lambda)$ and $a^*_d(\lambda)$ are, to some extent, ambiguous. Absorption of phytoplankton was compared with Chl*a* (excluding other pigments), and absorption of NAP (excluding phytoplankton) was only compared with the dry weight of all particles (including phytoplankton). As a result, our estimates of $a^*_{ph}(443)$ are probably higher than the actual $a_{ph}(443)$-to-phytoplankton dry weight ratio, and $a^*_d(443)$ is probably lower than the $a_d(443)$-to-NAP dry weight ratio [42]. The slope coefficient of NAP absorption ($S_d$) and CDOM absorption ($S_g$) was calculated by fitting an exponential equation over 400–700 nm with 440 nm as the reference band [43]. Further details on the field measurements of the bio-optical parameters and processing methods can be found in previous studies [31,32,44].

## 2.2. Sentinel-3A/OLCI Images

Sentinel-3A/OLCI Level-1B full-resolution data (OL_1_EFR, 300-m) were obtained from the European Space Agency (ESA) Copernicus Open Access Hub (https://scihub.copernicus.eu/dhus/#/home). A total of 101 cloud-free OLCI Level-1B images covering the lakes in the LYHR Basin, from 1 January 2017 to 31 December 2017, were collected. The 6SV atmospheric correction model (the vector version of the Second Simulation of the Satellite Signal in the Solar Spectrum correction scheme) [45] was applied to the cloud-free Level-1B OLCI images to acquire the OLCI-derived $R_{rs}(\lambda)$. The 6SV model was proven to be more efficient than other atmospheric correction methods in turbid inland waters [46]. A total of 63 match-up pairs of Sentinel-3A/OLCI data and field-measured data were acquired using a time window of ±3 h and a coefficient of variation (CV) test ($3 \times 3$ pixels, centered at the sampling station with CV <10%) [32,47]. Further details on the OLCI image preprocessing and validation of the performance of the atmospheric correction can be found in Shen, et al. [48].

$R_{rs}$ derived using C2RCC (Case 2 Regional Coast Color processor) [49], POLYMER (POLYnomial based algorithm applied to MERIS) [50], and 6SV were compared to field-measured $R_{rs}$ match-ups at 412, 443, 510, 560, 665, 681, 709, and 754 nm (Figure 2). 6SV was obviously superior (MAPE ranging from 14.78% to 57.93%) to C2RCC (52.21%–73.96%) and POLYMER (55.23%–91.37%) at the selected wavelengths. C2RCC and POLYMER tended to underestimate $R_{rs}$ in our dataset, while the 6SV also had relatively large uncertainties of $R_{rs}$ at 412 (MAPE = 57.93%), 443 (MAPE = 52.09%), and 754 nm (MAPE = 44.88%) (Figure 2d). The 6SV atmospheric correction method provided relatively accurate $R_{rs}$, and then 6SV-derived OLCI $R_{rs}$ was applied to optical classification and estimating bio-optical parameters.

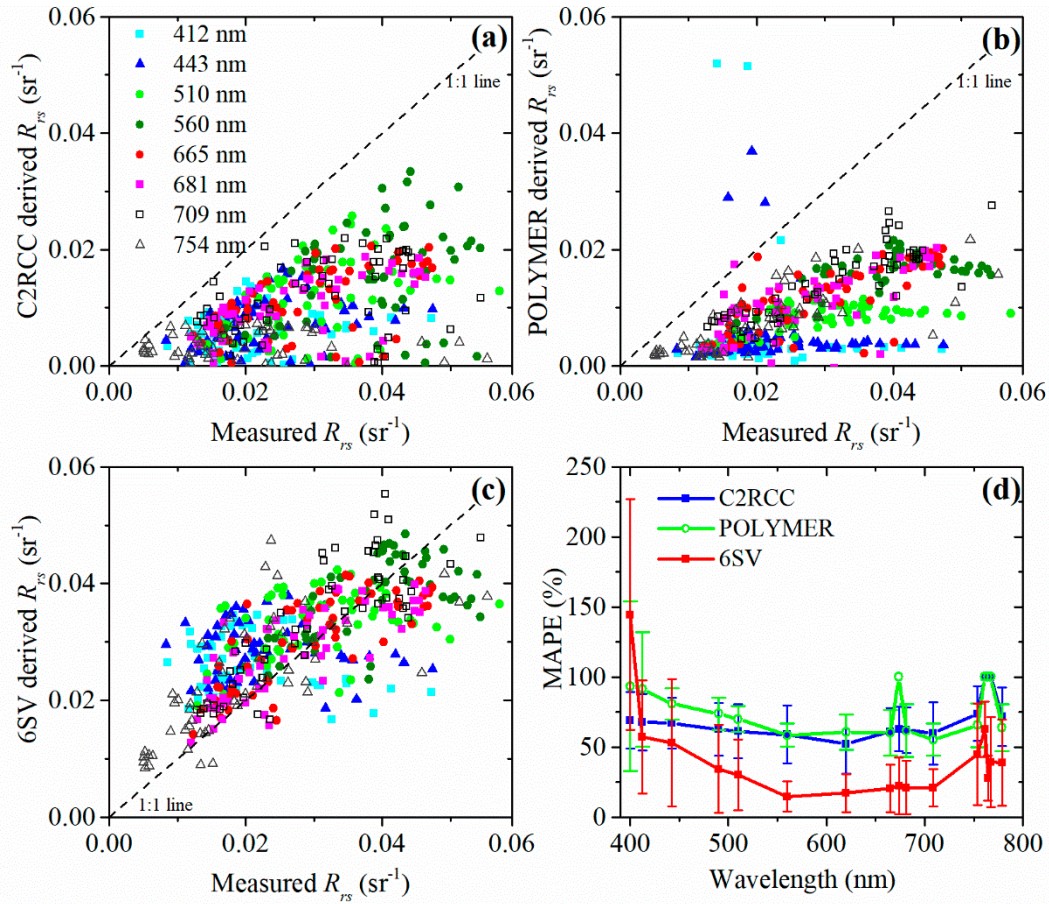

**Figure 2.** Comparison of the field-measured $R_{rs}$ and OLCI-derived $R_{rs}$ using the (**a**) C2RCC, (**b**) POLYMER, and (**c**) 6SV atmospheric correction models for match-up pairs at different OLCI bands ($N = 63$). (**d**) MAPE of C2RCC, POLYMER, and 6SV at different OLCI bands, error bars represent one standard deviation of the absolute percentage error in the validation data.

## 2.3. Optical Classification of the Remote Sensing Reflectance

### 2.3.1. Clustering the Optical Water Types Based on the Field $R_{rs}(\Lambda)$

The *k*-means classification approach was implemented into the normalized field-measured remote sensing reflectance [$NR_{rs}(\lambda)$, nm$^{-1}$] to generate the optical water types. Each $R_{rs}(\lambda)$ spectrum was normalized by its integrated value between 400 and 800 nm, similar to previous studies [11,19]. The equation of $NR_{rs}(\lambda)$ is as follows:

$$NR_{rs}(\lambda) = \frac{R_{rs}(\lambda)}{\int\limits_{400}^{800} R_{rs}(\lambda)\mathrm{d}\lambda}. \tag{1}$$

Each water type was defined by its average $NR_{rs}(\lambda)$ spectrum and covariance matrix from the $NR_{rs}(\lambda)$ spectra that belong to that water type in the clustering process. Other unsupervised clustering methods (e.g., heritage clustering, fuzzy *c*-means (FCM)) did not show better performance compared to *k*-means (Figure 3). The silhouette coefficient and SSE (sum of the squared errors) of the different number of types (from 2 to 10) were calculated and compared to determine the appropriate number of optical clusters. Heritage clustering and FCM did not change the average clustering curve dramatically [11], but had lower average silhouette coefficient and higher magnitude and variation of SSE and STD (standard deviation) (Figure 3). In addition, FCM is not effective in this study because we needed to derive distinctive clusters to understand the bio-optical properties of each OWT.

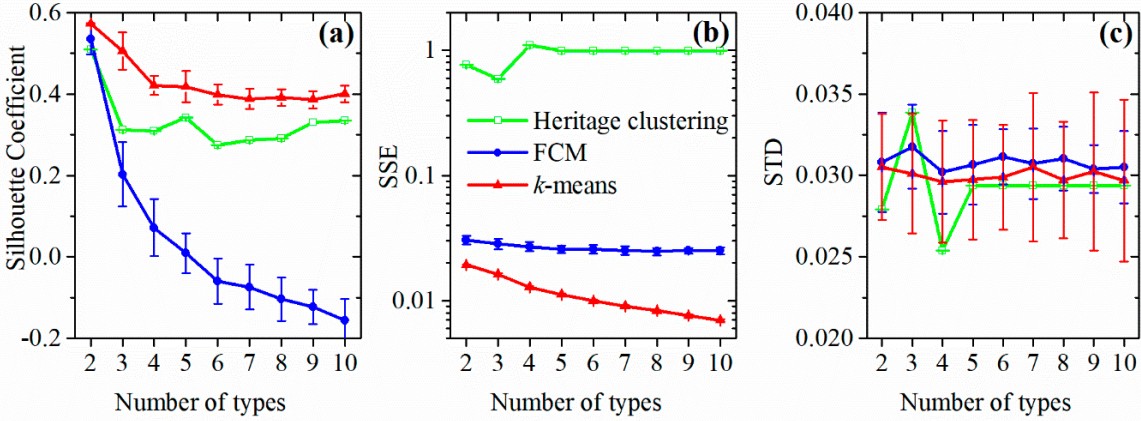

**Figure 3.** Performance of the three unsupervised clustering methods: heritage clustering, fuzzy *c*-means (FCM), and *k*-means in clustering waters with different number of types: (**a**) silhouette coefficient, (**b**) SSE (sum of the squared errors), and (**c**) STD (standard deviation).

### 2.3.2. Type-labeling of the Satellite $R_{rs}(\lambda)$

In this part, OLCI-derived $R_{rs}(\lambda)$ was associated with the different optical types identified from the field-measured data using *k*-means. Mahalanobis distance was used to identify the water type of satellite $R_{rs}(\lambda)$, and has good performance on ocean color data [6]. In this method, assuming for $R_{rs}(\lambda)$ a multivariate log-normal distribution of mean ($\mu$) and covariance matrix ($\Sigma$), the probability density function ($p$) associated with x = log($NR_{rs}$) was described as

$$p(x) = \frac{1}{(2\pi)^{d/2}|\Sigma|^{1/2}} \exp[-\frac{1}{2}(x-\mu)^T\Sigma^{-1}(x-\mu)], \tag{2}$$

where d is the dimension of $x$. The contours of constant probability associated with $p$ are defined by the related Mahalanobis distance ($D_m{}^2$) as follows:

$$D_m{}^2 = (x-\mu)^T\Sigma^{-1}(x-\mu). \tag{3}$$

Before labeling with the distinct type, the normalized OLCI $R_{rs}(\lambda)$ spectrum was log-transformed [6]. The $D_m{}^2$ of the input $NR_{rs}(\lambda)$ to a given water cluster was calculated, and used to determine the appropriate water type [11,19]. In addition, a theoretical threshold $D_t{}^2$, representing a given percentage (e.g., 90%) of the data distribution for a degree of freedom, was calculated according to the chi-square distribution. In this study, $D_t{}^2$ was 11.2, determined from the statistics of all the OLCI images. If $D_m{}^2$ was lower than the threshold value $D_t{}^2$, the spectrum x belongs to the class; if $D_m{}^2 > D_t{}^2$, the pixel would be recognized as an unclassified type. In addition, the current optical classification tool built in SNAP software [22] was also tried in this study, but it did not perform well due to the failure of atmospheric correction in the study region [48,51].

When applied to the satellite OLCI data, the wavelengths selected need to be effective in separate water types. The $NR_{rs}$ bands at 443, 490, 560, 620, 667, and 709 nm derived from the 6SV atmospheric correction were adopted in the class-matching of the OLCI images. The data at 400, 412, and 748 nm were not used due to the questionable accuracy of the atmospheric correction in inland waters [23]. The inclusion of $NR_{rs}(510)$ and $NR_{rs}(680)$ did not result in an improvement of the classification because of its strong correction to $NR_{rs}(490)$ and $NR_{rs}(667)$, respectively.

The Shannon index ($H$) of each pixel [52] was used to characterize the optical diversity of the waters from different OLCI images [19]:

$$H = -\sum_{i=1}^{NC} p'(i)\ln[p'(i)], \tag{4}$$

where *NC* is the number of classes, and *p′*(*i*) is the probability, representing the ratio between the number of images with type *i* and the number of all images for a given pixel. *H* has a maximum value (= ln(*NC*)) when the *NC* classes have the same probability. *H* is 0 if only one water type dominated with *p′* = 1.

### 2.4. Bio-Optical Algorithms Under Evaluation

Several Chl*a* inversion algorithms, including the NIR/red band-ratio algorithm (NR-2B) [53], 3-band algorithm developed for MERIS (Mer-3B) [36], fluorescence line height (FLH) [54], maximum chlorophyll index (MCI) algorithm [55], and EOF-based algorithm [56], have been developed in coastal and lake waters. Note that as it was not the intention of the study to develop a new index or algorithm for Chl*a* inversion, we focused on assessing the performance of the current algorithms in the optically classified waters.

The NR-2B algorithm uses the band ratio of NIR to red in a second-order polynomial equation:

$$\mathrm{NR}-2\mathrm{B} = \frac{R_{rs}(709)}{R_{rs}(665)}$$
$$\mathrm{Chl}a_{\mathrm{NR}-2\mathrm{B}} = A_1 \times \mathrm{NR}-2\mathrm{B}^2 + A_2 \times \mathrm{NR}-2\mathrm{B} + A_3 \tag{5}$$

The Mer-3B algorithm is expressed as follows:

$$\mathrm{Mer}-3\mathrm{B} = \left(\frac{1}{R_{rs}(665)} - \frac{1}{R_{rs}(709)}\right) \times R_{rs}(753)$$
$$\mathrm{Chl}a_{\mathrm{Mer}-3\mathrm{B}} = B_1 \times \mathrm{Mer}-3\mathrm{B} + B_2 \tag{6}$$

The MCI algorithm is expressed as follows:

$$\mathrm{MCI} = R_{rs}(709) - \left[R_{rs}(665) + (R_{rs}(754) - R_{rs}(665)) \times \frac{709-665}{754-665}\right]$$
$$\mathrm{Chl}a_{\mathrm{MCI}} = C_1 \times \exp(B_2 \times \mathrm{MCI}) \tag{7}$$

The parameters ($A_1$, $A_2$, $A_3$; $B_1$, $B_2$; and $C_1$, $C_2$) of the three Chl*a* estimation algorithms were first tuned using the overall field-measured data. After the optical classification, the three Chl*a* algorithms in each OWT were tuned by optimizing the parameters of each type using the corresponding field data of that type.

The relationship between $a^*_{ph}(\lambda)$ and Chl*a* can be represented by a power function [57]:

$$a^*_{ph}(\lambda) = A(\lambda) \times \mathrm{Chl}a^{-B(\lambda)}, \tag{8}$$

where $A(\lambda)$ and $B(\lambda)$ are positive, and represent the wavelength-dependent parameters in this relationship, which describes the decrease in $a^*_{ph}(\lambda)$ with increasing values of Chl*a*. The parameterization of $a^*_{ph}(\lambda)$ at 443 nm ($a^*_{ph}(443)$) was first performed based on the overall field $a^*_{ph}(443)$ and Chl*a* data. Then, the relationship between $a^*_{ph}(443)$ and Chl*a* of each water type was tuned using the data of that OWT. The combination of the class-based Chl*a* algorithm and the class-based $a^*_{ph}(443)$ algorithm was then applied to OLCI images to map $a^*_{ph}(443)$.

The mean absolute percentage error (MAPE) and the root mean square error (RMSE) between the field data ($X_i$) and the modeled data ($Y_i$) were calculated to evaluate the algorithm performance:

$$\mathrm{MAPE} = \frac{1}{n}\sum_{i=1}^{n} \frac{|Y_i - X_i|}{X_i} \times 100\%, \tag{9}$$

$$\mathrm{RMSE} = \sqrt{\frac{1}{n}\sum_{i=1}^{n}(\log 10(Y_i) - \log 10(X_i))^2} \tag{10}$$

The root mean squared difference (RMSD) was used to calculate the difference of two parameters derived from different methods (e.g., $R_{rs}$ derived using different values of $\varrho$):

$$\text{RMSD} = \sqrt{\frac{1}{n}\sum_{i=1}^{n}(X_{1,i} - X_{2,i})^2}. \tag{11}$$

## 3. Results

### 3.1. Optical Classification of the Remote Sensing Reflectance

Four OWTs were observed from the $NR_{rs}(\lambda)$ based on the *k*-means clustering method, and the order of the OWTs was changed according to the mean Chl*a* content value of each type (Figure 4). The majority of the samples were assigned to types 1–3 with percentages of 30%, 36%, and 31%, respectively. The $NR_{rs}(\lambda)$ of each OWT showed different magnitude and spectral characteristics. All types showed obvious peaks around 550, 650, and 700 nm. The overall differences between the four OWTs were the decreasing $NR_{rs}$ values in the blue to red range, and the increasing trend in the NIR range from type 1 to type 4. Type 1 had the lowest magnitude of $R_{rs}(\lambda)$ but had an obvious peak at approximately 550 nm. Type 2 and type 3 had higher values of $R_{rs}(\lambda)$. In addition, the $NR_{rs}(\lambda)$ of type 1 and type 2 overlapped from 570 to 620 nm, but had different magnitudes before and after this range. Type 3 showed relatively flat features compared to the other OWTs. Type 4 exhibited a comparable magnitude of $R_{rs}(\lambda)$ as type 1 in the blue to red range, but the highest value in the NIR range. The strong peak around 709 nm of type 4 indicated strong particle backscattering and was related to the high content of phytoplankton particles.

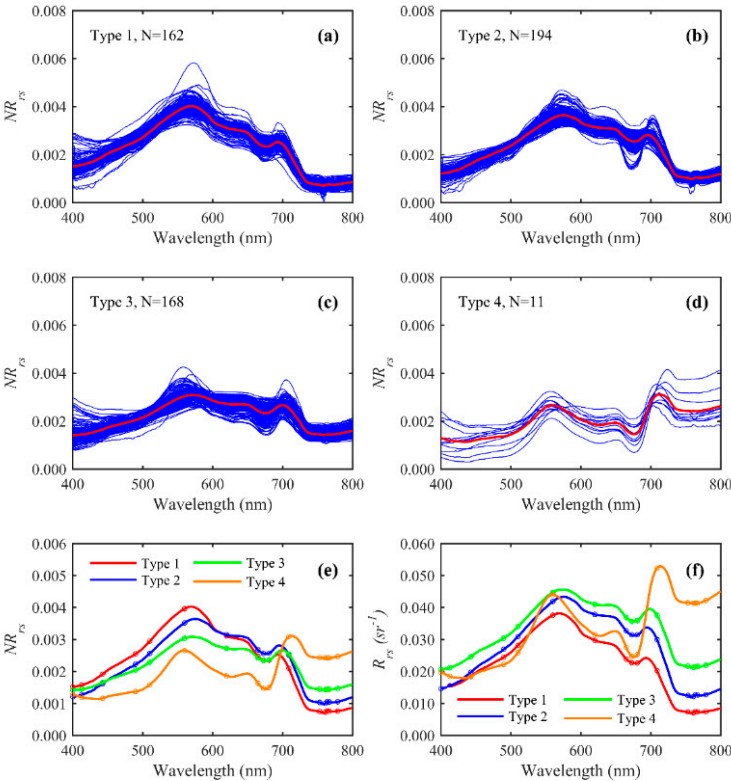

**Figure 4.** (**a**–**d**) $NR_{rs}(\lambda)$ sorted into the four optical water types (OWTs) from the *k*-means cluster analysis ($N$ = 535); blue lines: individual $NR_{rs}(\lambda)$ values; red lines: mean $NR_{rs}(\lambda)$ of each OWT. (**e**) The mean spectra of $NR_{rs}(\lambda)$ of the four OWTs. The OWT means and covariance matrices are the basis for the membership function. Note that the optical classification was conducted using the $NR_{rs}(\lambda)$ of the field data. (**f**) The mean spectra of $R_{rs}(\lambda)$ of the four OWTs.

## 3.2. Bio-Optical Characteristics of OWTs

Generally, the concentrations of water constitutes had a large range and variability in the overall dataset, with Chl*a* ranging from 0.70 to 382.03 mg/m$^3$, and SPM ranging from 5 to 245 g/m$^3$ (Table 1). After optical classification, types 1 through 4 had increasing mean Chl*a*, SPM, and SPOM contents (Table 1), and the corresponding mean values of the overall dataset were located between the mean values in type 2 and type 3. Type 1 had lowest mean Chl*a* and SPM magnitude and variability, indicating the clearest water among the four types. Type 4 had a notable high value of Chl*a* ($163.08 \pm 101.26$ mg/m$^3$) and a relatively higher mean SPIM ($47.88 \pm 29.78$ g/m$^3$) compared to the overall data ($37.13 \pm 27.16$ g/m$^3$).

The spectrum of $a_d(\lambda)$, $a_{ph}(\lambda)$, and $a_g(443)$ increased in magnitude from type 1 to type 4 (Figure 5). The peaks at approximately 443 and 675 nm are the common spectral characteristics of phytoplankton. The peak near 620 nm is the absorption peak of cyanobacteria, which was obvious in type 4. The $a_d(443)$ values were approximately two times those of $a_{ph}(443)$ in types 1–3; however, $a_{ph}(443)$ was notably larger than $a_d(443)$ in type 4 (Table 1). For the particulate absorption at 443 nm, type 2 had the lowest mean value ($0.27 \pm 0.15$) of $a_{ph}(443)/a_p(443)$, and type 4 had the highest value ($0.65 \pm 0.22$). $a_g(443)$ showed a relatively low mean value and variability compared with $a_{ph}(443)$ and $a_d(443)$ in each type. Type 1 had the lowest mean $a_g(443)$ ($0.78 \pm 0.45$ m$^{-1}$), and contributed more to $a(443)$ ($a_g(443)/a(443)$ = $0.25 \pm 0.11$) than other types. $a_g(443)$ had the highest mean value ($1.48 \pm 0.75$ m$^{-1}$) ranging from 0.73 to 3.18 in type 4. Overall, the variations in the content of the OACs and the associated absorption properties showed different features in each OWT, and the phytoplankton and inorganic particles dominated the optical variations in these waters.

**Table 1.** The mean value (mean $\pm$ SD) and range (min–max) of the field-measured concentrations of chlorophyll-*a* (Chl*a*, mg/m$^3$), suspended particulate matter (SPM, SPIM, and SPOM, g/m$^3$), total absorption coefficient at 443 nm ($a(443)$, m$^{-1}$), absorption coefficients of phytoplankton ($a_{ph}(443)$, m$^{-1}$), non-algal particles ($a_d(443)$, m$^{-1}$), and CDOM ($a_g(443)$, m$^{-1}$) at 443 nm, $a_{ph}(443)/a_p(443)$ ($a_p(443)$ = $a_{ph}(443) + a_d(443)$), and $a_g(443)/a(443)$. The "All" column contains the statistics of all data. Types 1–4 represent the statistics of each OWT.

| | All $N = 535$ | Type 1 $N = 162$ | Type 2 $N = 194$ | Type 3 $N = 168$ | Type 4 $N = 11$ |
|---|---|---|---|---|---|
| Chl*a* | $31.77 \pm 36.86$ | $19.30 \pm 13.57$ | $26.56 \pm 25.56$ | $41.47 \pm 37.21$ | $163.08 \pm 101.26$ |
| | 0.70–382.03 | 1.27–85.64 | 0.70–165.84 | 0.71–157.05 | 70.41–382.03 |
| SPM | $48.87 \pm 30.31$ | $30.37 \pm 11.48$ | $45.07 \pm 20.23$ | $68.85 \pm 36.88$ | $91.82 \pm 45.22$ |
| | 5.00–245.00 | 5.00–73.33 | 5.00–150.00 | 10.67–245.00 | 20.00–210.67 |
| SPIM | $37.13 \pm 27.16$ | $21.44 \pm 12.67$ | $37.86 \pm 18.67$ | $50.91 \pm 36.39$ | $47.88 \pm 29.78$ |
| | 0.50–232.00 | 0.50–73.00 | 6.00–110.00 | 4.00–232.00 | 1.33–96.00 |
| SPOM | $16.77 \pm 16.04$ | $12.18 \pm 7.35$ | $15.23 \pm 12.99$ | $20.63 \pm 18.27$ | $51.12 \pm 44.11$ |
| | 1.00–173.33 | 2.67–50.00 | 1.00–120.00 | 1.00–107.00 | 16.00–173.33 |
| $a(443)$ | $4.67 \pm 2.25$ | $3.15 \pm 0.80$ | $4.49 \pm 1.22$ | $5.96 \pm 2.32$ | $11.27 \pm 5.13$ |
| | 1.02–20.86 | 1.02–5.61 | 2.06–11.41 | 2.24–16.18 | 5.34–20.86 |
| $a_{ph}(443)$ | $1.31 \pm 1.56$ | $0.86 \pm 0.46$ | $0.99 \pm 0.80$ | $1.76 \pm 1.63$ | $6.91 \pm 5.17$ |
| | 0.16–17.88 | 0.16–3.09 | 0.18–5.50 | 0.20–13.12 | 1.80–17.88 |
| $a_d(443)$ | $2.40 \pm 1.37$ | $1.52 \pm 0.58$ | $2.49 \pm 0.80$ | $3.11 \pm 1.86$ | $2.88 \pm 1.84$ |
| | 0.34–10.41 | 0.34–2.97 | 0.51–5.50 | 0.39–10.41 | 0.59–5.66 |
| $a_g(443)$ | $0.98 \pm 0.60$ | $0.78 \pm 0.45$ | $1.02 \pm 0.70$ | $1.10 \pm 0.51$ | $1.48 \pm 0.75$ |
| | 0.16–7.10 | 0.16–2.50 | 0.28–7.10 | 0.28–4.04 | 0.73–3.18 |
| $a_{ph}(443)/a_p(443)$ | $0.34 \pm 0.18$ | $0.36 \pm 0.15$ | $0.27 \pm 0.15$ | $0.36 \pm 0.21$ | $0.65 \pm 0.22$ |
| | 0.06–0.97 | 0.13–0.76 | 0.06–0.83 | 0.07–0.92 | 0.31–0.97 |
| $a_g(443)/a(443)$ | $0.22 \pm 0.11$ | $0.25 \pm 0.11$ | $0.23 \pm 0.10$ | $0.21 \pm 0.10$ | $0.15 \pm 0.08$ |
| | 0.05–0.62 | 0.05–0.60 | 0.06–0.62 | 0.05–0.55 | 0.06–0.37 |

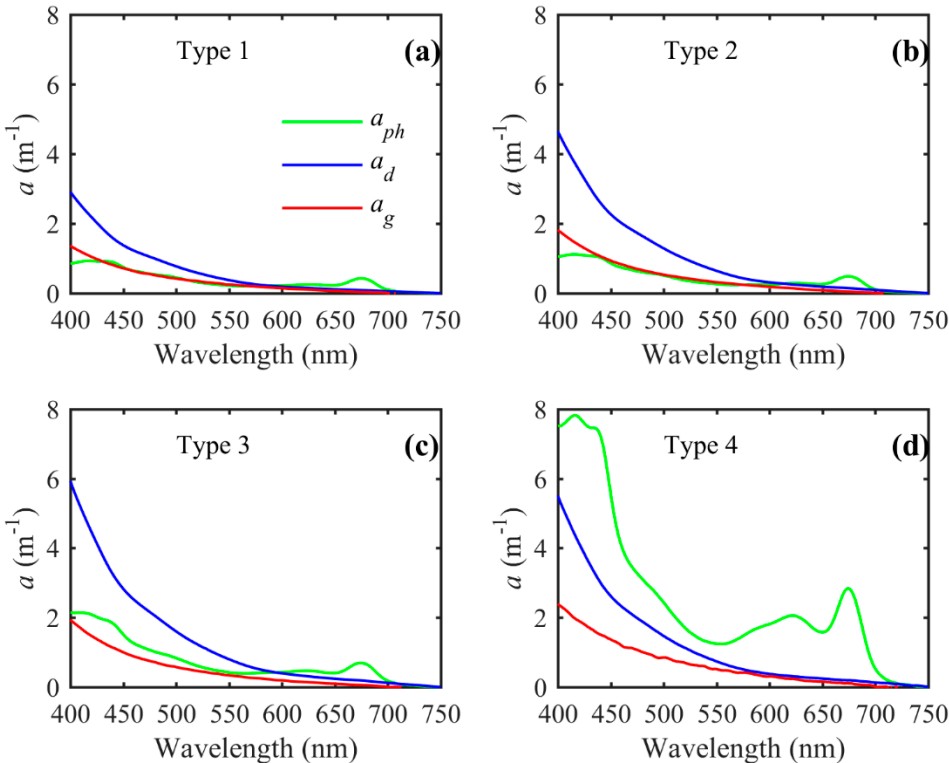

**Figure 5.** Mean spectrum of the absorption coefficients of phytoplankton ($a_{ph}$), NAP ($a_d$), and CDOM ($a_g$) in each OWT: (**a**) type 1, (**b**) type 2, (**c**) type 3, and (**d**) type 4.

The average spectrum of $a_{ph}^*(\lambda)$ of each OWT showed that type 1 and type 2 had similar magnitudes, while type 3 had the highest and type 4 had the lowest values in the range from 400 to 650 nm (Figure 6a). Type 1 had a relatively higher $a_{ph}^*(675)$ (0.024 m$^2$/mg) value compared to the other OWTs (0.021, 0.020, 0.017 m$^2$/mg for types 2–4, respectively), indicating a larger proportion of small cells in type 1. Types 1–3 showed increasing mean values of $a_{ph}^*(443)/a_{ph}^*(675)$, which indicated the effects of accessory pigments on the variation in the phytoplankton absorption. The highest mean value of $a_{ph}^*(443)/a_{ph}^*(675)$ in type 3 was related to the high influence of the accessory pigments. The low magnitude of $a_{ph}^*(443)$ in type 4 (0.042 ± 0.0003 m$^2$/mg) was mainly affected by the packaging effect in algal bloom waters with accumulated algae.

The mean $a_d^*(\lambda)$ spectra of types 1–3 were very similar, and type 2 (0.066 m$^2$/g) had a slightly higher value compared to type 1 (0.054 m$^2$/g) and type 3 (0.050 m$^2$/g). However, $a_d^*(443)$ of type 2 was approximately 2 times that of type 4 (0.033 m$^2$/g), resulting from the higher proportion of NAP in type 2 compared to type 4. In addition, $S_d$ and $S_g$ did not show significant differences ($p > 0.5$), with average values of 0.0112 and 0.0105, respectively.

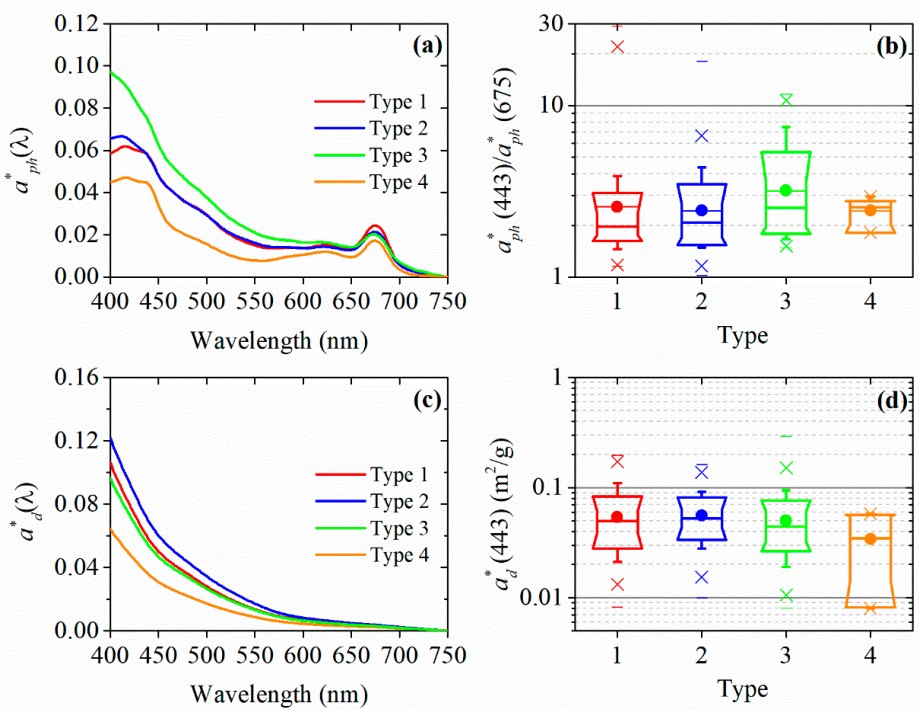

**Figure 6.** (**a**) Mean spectra of the absorption coefficient of phytoplankton normalized to the Chl*a* concentration ($a^*_{ph}(\lambda)$) of types 1–4. (**b**) Boxplots of $a^*_{ph}(443)/a^*_{ph}(675)$ for each OWT in the field-measured data. (**c**) Mean spectra of the absorption coefficient of NAP normalized to the SPM concentration ($a^*_d(\lambda)$) of types 1–4. (**d**) Boxplots of $a^*_d(443)$ for each OWT in the field-measured data. The sample median is indicated by a line within the box, the dots represent the mean value, and "x" represents data beyond the bounds of the error bars.

### 3.3. Application to the $a_{ph}^*(443)$ Estimation

#### 3.3.1. Model Validation

The performance of the Chl*a* algorithms using the three indexes (NR-2B, Mer-3B, and MCI) was assessed using the field dataset (Table 2). The RMSE of the three models based on the overall data were similar (21.78, 28.57, and 23.35 mg/m$^3$ for NR-2B, Mer-3B, and MCI, respectively), while the MAPE of Mer-3B (104.90%) was higher than those of NR-2B (71.34%) and MCI (56.23%). The performance of the class-specific Chl*a* models in the individual OWTs indicated that type 1 and type 2 had an apparent improvement in the three Chl*a* models. Mer-3B performed better than NR-2B and MCI, with a lower RMSE and APD in type 1. Type 3 had similar RMSE values compared to the three algorithms, while type 4 had obviously larger RMSEs compared to the overall dataset. Comparably, the waters with an optical classification except type 4 showed an improved performance in the class-specific NR-2B and Mer-3B Chl*a* models.

**Table 2.** Uncertainty statistics of the four OWTs and all data for the derived Chl*a* from the three algorithms (NR-2B, Mer-3B, and MCI) using the field-measured data.

|  | NR-2B | | | Mer-3B | | | MCI | | |
|---|---|---|---|---|---|---|---|---|---|
|  | $R^2$ | RMSE (mg/m$^3$) | MAPE (%) | $R^2$ | RMSE (mg/m$^3$) | MAPE (%) | $R^2$ | RMSE (mg/m$^3$) | MAPE (%) |
| Type 1 | 0.53 | 9.30 | 40.53 | 0.66 | 7.32 | 34.19 | 0.35 | 10.93 | 47.04 |
| Type 2 | 0.86 | 9.70 | 39.52 | 0.88 | 9.79 | 40.33 | 0.63 | 15.37 | 53.60 |
| Type 3 | 0.63 | 23.35 | 68.26 | 0.64 | 22.99 | 59.12 | 0.60 | 25.03 | 69.11 |
| Type 4 | 0.18 | 87.59 | 42.91 | 0.01 | 96.13 | 51.70 | 0.07 | 92.92 | 47.24 |
| All data | 0.66 | 21.78 | 71.34 | 0.51 | 28.57 | 104.90 | 0.61 | 23.35 | 56.23 |

The relationships between Mer-3B and Chl$a$ of each OWT were then established using the match-up pairs of field-measured data and OLCI-derived data (Figure 7a). The Chl$a$ exhibited an increasing trend with increasing Mer-3B; however, the function of the overall data (black line) indicated a clear overestimation when Chl$a$ < 10 mg/m$^3$. In addition, the relationships between the Chl$a$ and $a_{ph}^*$(443) of each OWT and the overall data ($a_{ph}^*$(443) = $n_1*$Chl$a^{-n2}$) were also developed using the match-up pairs (Figure 7b). $a_{ph}^*$(443) was well correlated with the Chl$a$ content in types 1–2, while $a_{ph}^*$(443) did not show a good relationship with Chl$a$ in type 4. Thus, $a_{ph}^*$(443) of type 4 was calculated using the function of the overall data.

The comparison between the field-measured and model-derived Chl$a$ content indicated the improvement of Chl$a$ estimation using the class-specific algorithms of the different OWTs (Figure 7c). In particular, the RMSE of deriving Chl$a$ decreased from 19.01 and 13.77 mg/m$^3$ to 12.37 and 9.98 mg/m$^3$ in types 1 and 2, respectively (Figure 7c). The estimation of $a_{ph}^*$(443) showed an obvious improvement in types 1 and 2 using the class-specific $a_{ph}^*$(443) model of each OWT. The combination of class-specific Chl$a$ algorithms (Figure 7a) and class-specific $a_{ph}^*$(443) algorithms (Figure 7b) could provide an effective way to estimate $a_{ph}^*$(443) in waters with large optical variations.

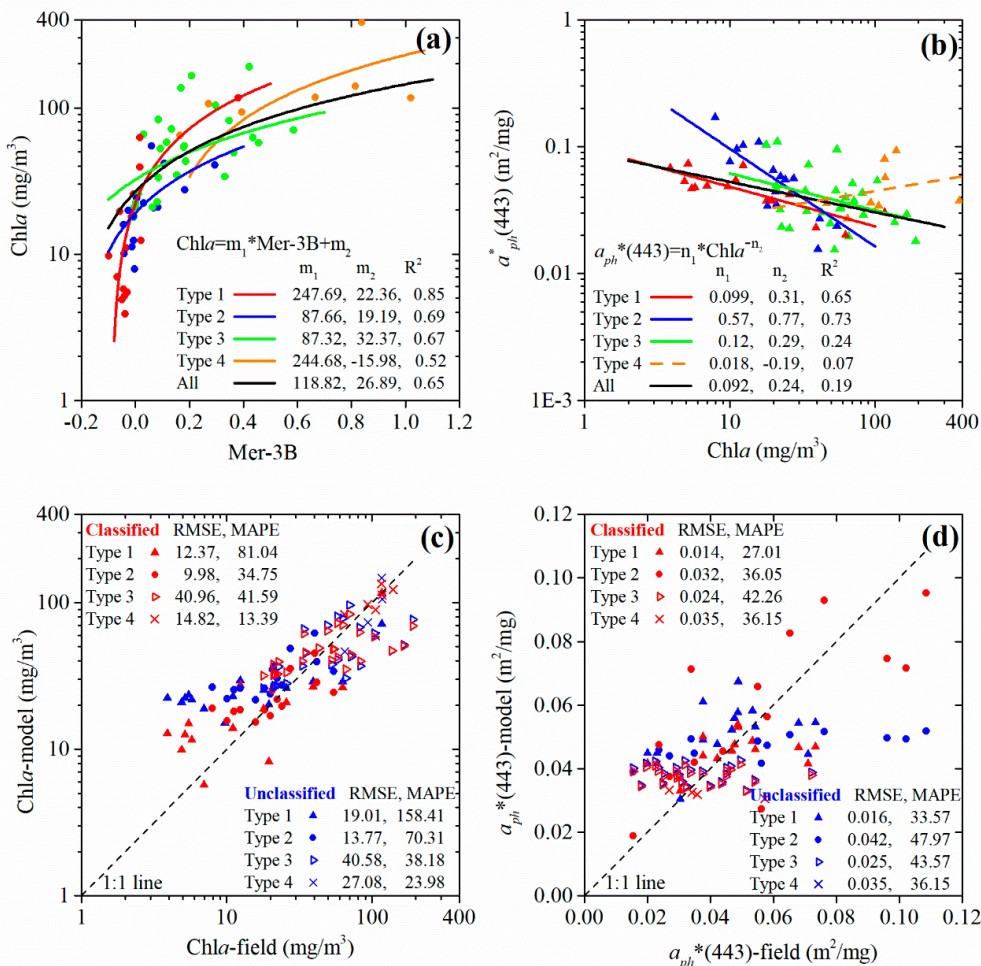

**Figure 7.** (**a**) Mer-3B versus field-measured Chl$a$ content data for OLCI validation of each OWT and all data. (**b**) Chl$a$ versus $a_{ph}^*$(443) for OLCI validation of each OWT and all data. (**c**) Comparison of the field-measured Chl$a$ and model-derived Chl$a$ using unclassified models and classified models for each OWT and all data. (**d**) Comparison of the field-measured $a_{ph}^*$(443) and model-derived $a_{ph}^*$(443) using unclassified models and classified models for each OWT and all data. Note that the input Chl$a$ data in calculating $a_{ph}^*$(443) were the derived Chl$a$ values using the class-specific model of each OWT. The number of samples (*N*) is 15, 15, 27, and 6, for type 1 to type 4, respectively.

### 3.3.2. Application to the Satellite OLCI Data

The optical classification method was then applied to the OLCI-derived $NR_{rs}(\lambda)$ to map the water types of the lakes in the LYHR Basin on 2 March 2017 and 24 October 2017 (Figure 8a,e). The dominant OWTs were type 2 and type 3 on 2 March 2017 (Figure 8a), while the dominant OWTs were type 1 and type 2 on 24 October 2017 (Figure 8e). The black regions are the areas that were not classified as any water type based on the classification criteria, due to the cloud coverage, land adjacency, or aquatic vegetation in the lakes. Large lakes, such as Lake Taihu, Lake Hongze, and Lake Chaohu, were usually dominated by types 1 and 2. Type 4 was located in the northern part of Lake Taihu on 24 October 2017, due to the occurrence of algal blooms. Furthermore, the class-specific Chl*a* and $a_{ph}^{*}(443)$ algorithms were used to derive the corresponding Chl*a* content and $a_{ph}^{*}(443)$ in each OWT. Compared with Chl*a* derived using the unclassified Mer-3B model (Figure 8b,f), a large range of Chl*a* values was derived with the class-specific Mer-3B Chl*a* algorithm, which improved the performance of Chl*a* estimation at low values. Then, $a_{ph}^{*}(443)$ was derived using the classified $a_{ph}^{*}(443)$ models based on the class-specific model-derived Chl*a*. Overall, $a_{ph}^{*}(443)$ had an inverse tendency with the Chl*a* distribution. The central part of Lake Hongze and the western part of Lake Taihu had high $a_{ph}^{*}(443)$ values on 2 March 2017.

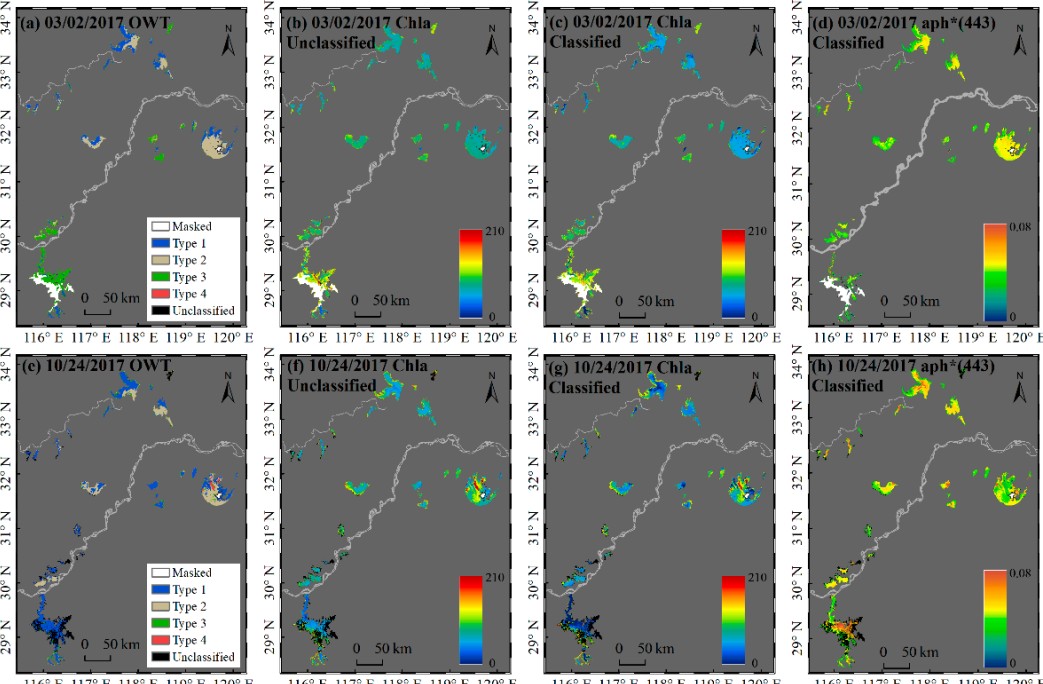

**Figure 8.** (**a**) Optical water types, (**b**) Chl*a* derived using the unclassified Mer-3B Chl*a* model, (**c**) Chl*a* derived using the class-specific Mer-3B Chl*a* model, and (**d**) $a_{ph}^{*}(443)$ derived using the class-specific model on the 2 March 2017, OLCI image over the lakes in the LYHR Basin. (**e**) Optical water types, (**f**) Chl*a* derived using the unclassified Mer-3B Chl*a* model, (**g**) Chl*a* derived using the class-specific Mer-3B Chl*a* model, and (**h**) $a_{ph}^{*}(443)$ derived using the class-specific model on the 24 October 2017, OLCI image over the lakes in the LYHR Basin.

## 4. Discussion

Optical classification is an effective way to distinguish optical water types in oceanic, coastal, and lake waters. Different from the true end-member classes in the land cover classification scheme, the optical water types are determined from the characteristics of $R_{rs}(\lambda)$ or $L_{w}(\lambda)$, and reflect the optical conditions of the water, which could change dramatically with time [23]. The choice of the optical classification scheme depends on the usage, e.g., to determine the most suitable tuning method of a bio-optical algorithm, or to assess the general optical conditions of the lakes [16,22]. In optical

classification, the $R_{rs}(\lambda)$ spectra and normalized $R_{rs}(\lambda)$ spectra were both used to define the OWTs. The variability in the magnitude of $R_{rs}(\lambda)$ is mostly associated with backscattering and concentration of particles, whereas the absorption coefficients of each component are more related to the spectral shape [19,58]. That is, the optical classification based on the normalized $R_{rs}(\lambda)$ focused on spectral shape variations, whereas the optical classification based on $R_{rs}(\lambda)$ is greatly influenced by the gradient in the concentrations of SPM.

The appropriate number of clusters is usually determined prior to using different methods, including gap statistic [16] and cluster validity measures [21], and is adjusted automatically based on the spectral standard deviation and distance criteria [19]. In this study, the appropriate number of clusters was determined using gap statistics [16]. The number of water types was similar to the previous studies in Lake Taihu and Lake Chaohu, which illustrated three water types using a hierarchical approach [15] and the TD680 water classification method [30]. In addition, the selection of wavebands in the type-labeling of the satellite $R_{rs}$ also affected the effectiveness of the optical classification. Note that the covariance matrix would increase as the square of the number of labeling wavelengths [59]. Similar to the previous study [23], $R_{rs}(400)$, $R_{rs}(412)$, and $R_{rs}$ of NIR bands longer than 709 nm were omitted in the classification due to the poor performance of the atmospheric correction. For OLCI-derived $R_{rs}$, POLYMER and C2RCC had obvious overcorrection of $R_{rs}$, consistent with the study of Bi et al. (2018) [51] in Lake Taihu and Lake Hongze. It was also shown that C2RCC exhibited good performances from 490 to 709 nm, and poor performances in the blue (400, 412, and 443 nm) and NIR wavebands (754–865 nm) for the highly absorbing waters in the Baltic Sea [60]. However, 6SV had better performance than POLYMER and C2RCC in the turbid and eutrophic waters in this study.

One limitation of defining the optical classes using the field $R_{rs}(\lambda)$ data is that the optical variability in the OWTs is restricted to the range of the field data. If there exists a water type that was not included or only represented a small fraction of the field data, the results would be unclassified or classified into a similar water type [19]. As we would like to analyze the bio-optical properties and build class-specific models, the optical classification based on field data was necessary. Jackson et al. suggested that optical classification on a global scale can be first used to highlight regions where more sampling would be of great significance [59]. Several studies [16,23,61] have provided valuable frameworks for classifying global waters; however, the OWTs from the large global dataset cannot be used in regional studies of inland lakes. Figure 9 shows that the OWTs in this study are different from those illustrated in Table A1 in Moore et al. (2009) and Table 2 in Moore et al. (2014) [21,23] and located between type 6 and type 7 of Moore et al. (2014). Type 8 in Moore et al. (2009) had higher $R_{rs}(\lambda)$ values in the blue band and lower $R_{rs}(\lambda)$ values in the red band, compared to the OWTs in this study. The latter could explain the reason that the optical classification using the approach in Moore et al. (2009; 2014) did not obtain suitable results (data not shown). This finding indicated not only the difficulty of using the OWTs of other studies directly, but also the importance of considering the usage of optical classification. If optical classification is used to characterize the optical conditions of global or large-scale waters, coarse water types may be suitable. A finer optical classification is suggested in developing class-specific or blended inversion models, which could provide more reliable results.

The optical variations in the lakes in the LYHR Basin in 2017 were illustrated using the dominant OWT and Shannon index (*H*) (Figure 10). Type 1 dominated most of the lakes through 2017, type 2 dominated the southern part of Lake Taihu, and type 4 dominated the western part of Lake Taihu. *H*, ranging from 0 to 1.4, indicated the optical similarity and diversity of the lakes in 2017 (Figure 10b). Most of the lakes had *H* values between 0.5 and 1.2, with an average value of 0.84 ± 0.05. The northern part of Lake Hongze had a low *H* value, while the western and southern part of Lake Taihu, the southern part of Lake Hongze, Lake Chaohu, and several small lakes near the Yangtze River had high *H* values, indicating the optical diversity in these areas. In addition, the frequency of each OWT in 2017 showed that type 1 and type 2 contributed most of the percentage except for areas with turbid waters (type 3) and algal blooms (type 4). The northern part of Lake Hongze was dominated by type 1, and the southern part of Lake Hongze was dominated by types 1 and 2. However,

the northwestern part of Lake Taihu and the northwestern part of Lake Chaohu also contributed to type 4, indicating the frequent occurrence of algal blooms.

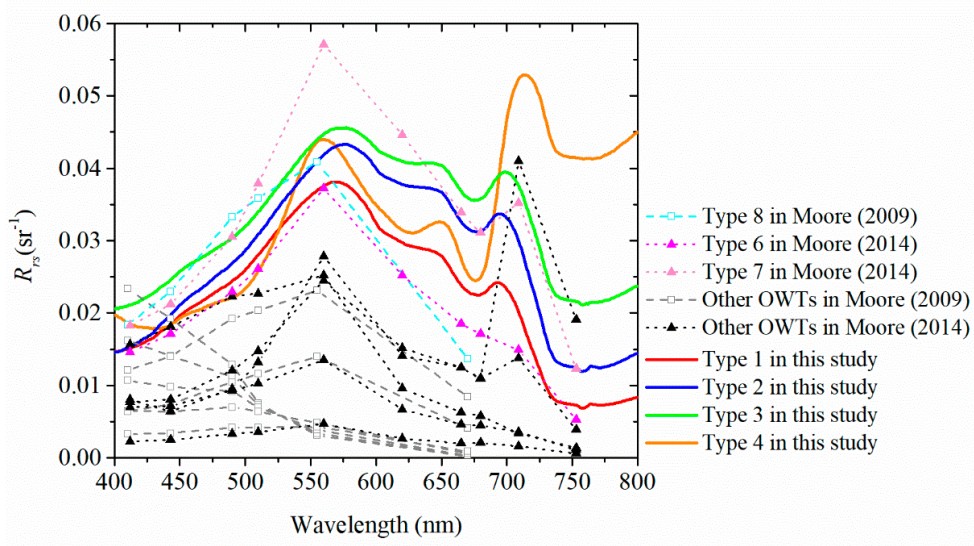

**Figure 9.** The comparison of mean $R_{rs}(\lambda)$ of the four optical water types with the optical water types in the previous studies [21,23]. The dashed lines represent mean $R_{rs}(\lambda)$ of OWTs acquired from Table A1 in Moore et al. (2009) [21] and Table 2 in Moore et al. (2014) [23].

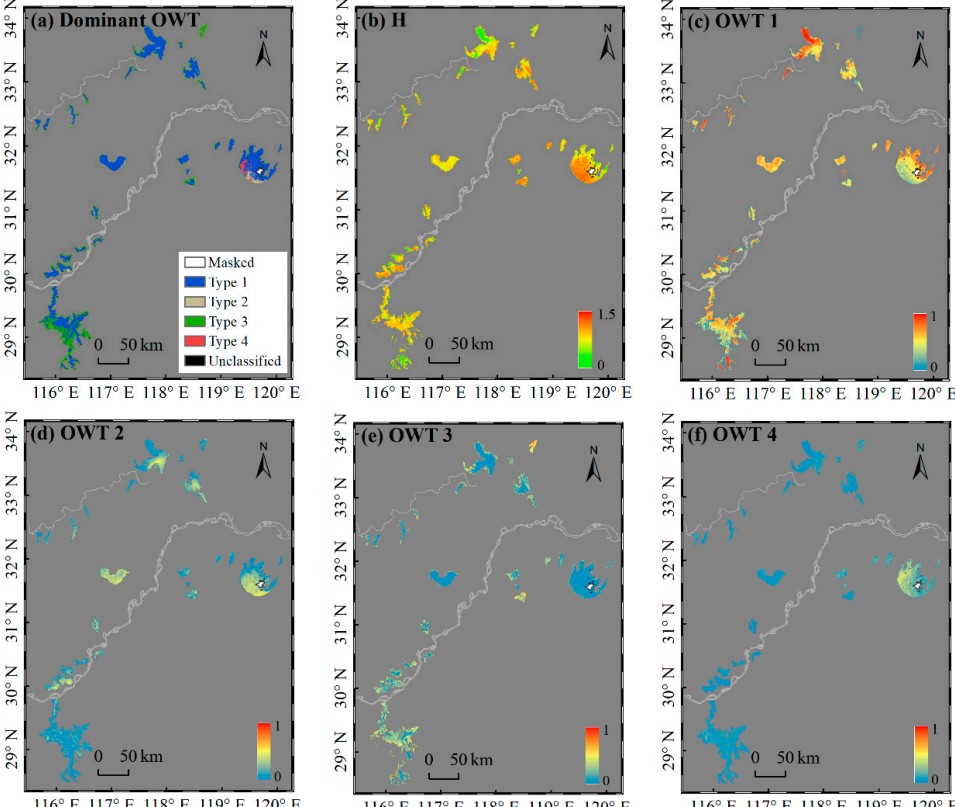

**Figure 10.** (**a**) Dominant OWTs of the lakes in the LYHR Basin in 2017 (the class most frequently selected as the dominant class over the period); (**b**) Shannon index (*H*) computed from the frequency of the different OWTs of the lakes in the LYHR Basin in 2017. (**c–f**) The annual frequency of the different OWTs: (**c**) type 1, (**d**) type 2, (**e**) type 3, (**f**) type 4, associated with lakes in the LYHR basin in 2017.

The western part of Lake Chaohu and the western part of Lake Taihu had high optical diversity, which is in accordance with previous studies [56,62]. The mean field-measured Chl*a* content value in this study was $31.77 \pm 36.86$ mg/m$^3$, ranging from 0.7 to 382.03 mg/m$^3$. The mean value of Chl*a* was 40.5 mg/m$^3$, ranging from 4.0 to 448.9 mg/m$^3$ in Lake Taihu in the period of 2006 to 2007 [62], and the western lake and Meiliang Bay of Lake Taihu had a high variation in the Chl*a* content [56]. The western part of Lake Chaohu showed the highest Chl*a* through the seasonal cycle (21.96–63.63 mg/m$^3$), followed by the eastern part (19.26–54.95 mg/m$^3$) and the central part (17.31–51.87 mg/m$^3$) of Lake Chaohu [2].

The relations between $a^*_{ph}(\lambda)$ and Chl*a* have been used in estimating $a^*_{ph}(\lambda)$ and in modeling of the primary production [57,63]. The variation in $a^*_{ph}(\lambda)$ was usually affected by the package effect and accessory pigments, which resulted in the weak correlation between $a^*_{ph}(\lambda)$ and Chl*a*. Relatively low $a^*_{ph}(\lambda)$ values and an independence of $a^*_{ph}(\lambda)$ with regards to Chl*a* were usually observed in the highly eutrophic waters [64,65]. In this study, the mean $a^*_{ph}(675)$ values of each OWT ($0.024 \pm 0.001$, $0.021 \pm 0.001$, $0.021 \pm 0.001$, and $0.017 \pm 0.001$ m$^2$ mg$^{-1}$ for types 1–4, respectively) were compared with the values in the previous studies of high eutrophic lakes, e.g., Lake Taihu ($0.021 \pm 0.011$ m$^2$ mg$^{-1}$ [64], and 0.022 m$^2$ mg$^{-1}$ [66]) and Lake Kasumigaura ($0.018 \pm 0.005$ m$^2$ mg$^{-1}$) [65]. The high $a^*_{ph}(675)$ value in type 1 indicated the high content of small cells. The low mean value of $a^*_{ph}(675)$ and its poor relationship with Chl*a* content values were also observed in the waters of type 4 ($0.017 \pm 0.001$ m$^2$mg$^{-1}$). Note that after the optical classification, $a^*_{ph}(675)$ had a low variability in each OWT compared to the $a^*_{ph}(675)$ of the overall data ($0.022 \pm 0.011$ m$^2$ mg$^{-1}$).

The uncertainties in input field-measured $R_{rs}(\lambda)$ affect the accuracy of optical classification and bio-optical models [41,67]. In the measurement of $R_{rs}(\lambda)$ using above-water approach, water-leaving radiance ($L_w(\lambda)$) is derived by correcting the measured above-water upwelling radiance ($L_u(\lambda)$) using a reflectance ratio ($\varrho$) which depends on sky conditions, wind speed, solar zenith angle [34], sky polarization [35], and wavelength [68]. According to the look-up table of Mobley [34] (M1999) and measurement conditions, $\varrho = 0.028$ was used in this study. For the concurrent validation data ($N = 63$), comparison of $R_{rs}(\lambda)$ derived using $\varrho$ in Mobley (2015) ($R_{rs\text{-}M2015}(\lambda)$), $\varrho$ in Mobley (1999) ($R_{rs\text{-}M1999}(\lambda)$), and $\varrho = 0.028$ indicated that $\varrho = 0.028$ had lower RMSD than that of M1999, especially in the wavelength range > 500 nm (Figure 11). Band combination in NR-2B and Mer-3B could decrease this variability introducing from surface-reflected light (Figure 11c). In addition, it was demonstrated that there is no general value of $\varrho$ to be adopted in different inland water conditions, but the most suitable methodology is the spectral $\varrho$, e.g., approach in Lee et al. (2010) [69]. However, spectral variability of $\varrho$ was not taken into consideration in the process of deriving field-measured $R_{rs}(\lambda)$, which should be improved in the further studies.

The main aims of this study were to document the optical variations in the lakes in the LYHR Basin and to refine the bio-optical algorithms through optical classification. The specific absorption coefficients, especially the Chl*a*-specific phytoplankton coefficients of the four OWTs, had significant differences. This finding indicated that the variations in the specific inherent optical properties (SIOPs) should be taken into consideration in establishing bio-optical inversion models in waters with different OWTs. Moreover, the optical similarity and variability usually reflect the optical conditions and can be used in the selection of algorithms for specific regions [19]. The bio-optical inversion models mostly had certain limitations and a range of applicability. It is more likely that a model can be used for other waters with a high degree of optical similarity. In addition, the SIOPs of each OWT can be used as input parameters of radiative transfer simulation, e.g., HydroLight, Monte Carlo simulation, in studying the underwater light field and light fluctuations in optically dynamic waters.

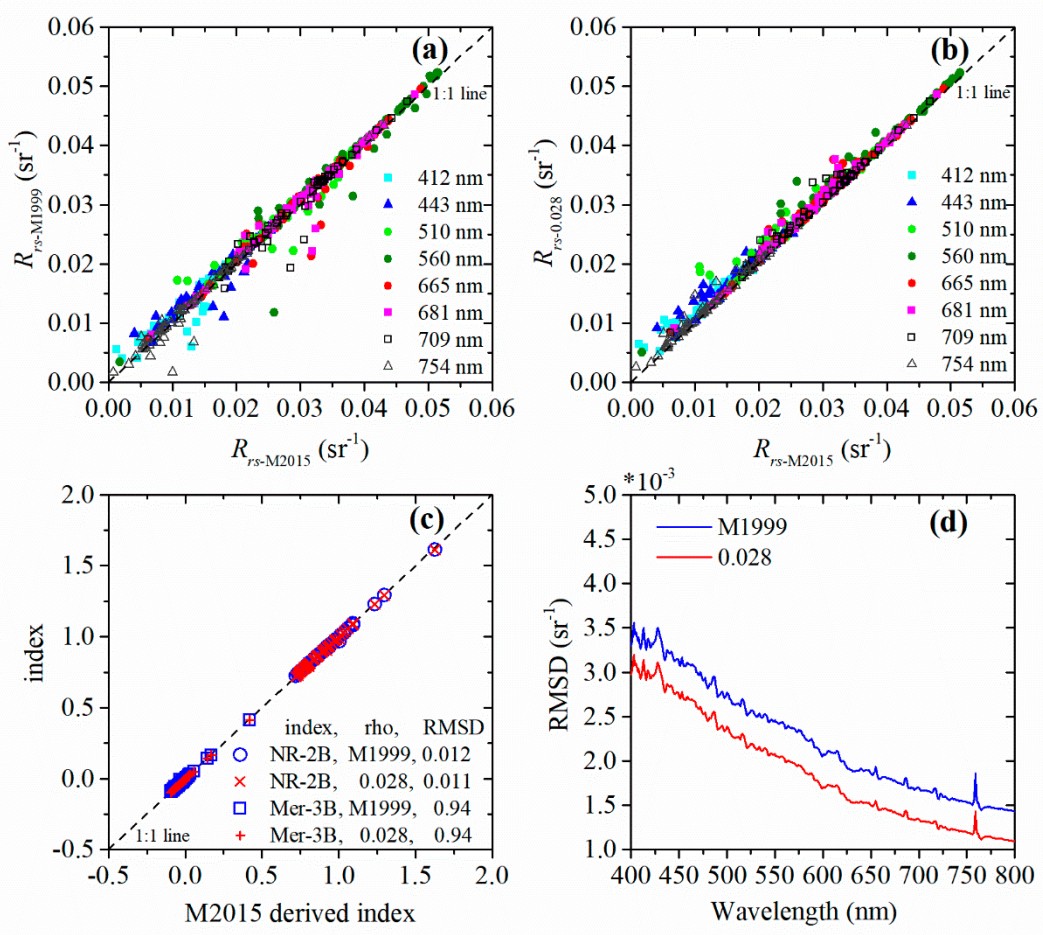

**Figure 11.** Comparison of $R_{rs}(\lambda)$ derived using $\varrho$ in Mobley (2015) [35] ($R_{rs\text{-}M2015}(\lambda)$) and (**a**) $R_{rs}(\lambda)$ derived using $\varrho$ in Mobley (1999) [34] ($R_{rs\text{-}M1999}(\lambda)$), and (**b**) using $\varrho = 0.028$ for match-up pairs ($N = 63$). (**c**) Comparisons between indexes (NR-2B, Mer-3B) derived using M2015 and M1999, 0.028, respectively. (**d**) Spectral RMSD of $R_{rs}(\lambda)$ between $\varrho$ of M2015 and M1999 (blue line), 0.028 (red line), respectively.

## 5. Conclusion

Optical classification was used to characterize the optical variations and evaluate the potential of estimating $a^{*}_{ph}(443)$ of the lakes in the LYHR Basin. Four OWTs were derived using $NR_{rs}(\lambda)$, and the bio-optical properties of each OWT were compared. Type 2 showed an obvious feature with a high contribution of mineral particles, while type 4 was mostly determined by a high content of phytoplankton. The $a_{g}(443)$ values did not show significant differences among the 4 water types. Furthermore, the potential of class-specific inversion algorithms for estimating $a^{*}_{ph}(443)$ was illustrated by developing class-specific Chl*a* inversion algorithms first. An improved performance of the class-specific algorithms was demonstrated in each optical water type, especially in types 1–2. In addition, the optical variation in and similarity of the lakes in the LYHR Basin were characterized using the dominant water type and Shannon index (*H*), respectively, in 2017. A high optical variation was located in the western and southern parts of Lake Taihu, the southern part of Lake Hongze, Lake Chaohu, and several small lakes near the Yangtze River, while the northern part of Lake Hongze had a low optical diversity. The results indicated the necessity of optical classification in lakes with a large range and variability in the bio-optical parameters. The class-specific inversion algorithms for estimating the bio-optical parameters are suitable for waters in optically complex and dynamic lakes. In the future, analysis of the temporal variations in the water types would help towards understanding the influence of ecological processes and environmental conditions on the spatial-temporal variations in bio-optical parameters.

**Author Contributions:** Conceptualization, R.M.; Data curation, K.X. and D.W.; Formal analysis, K.X., M.S.; Funding acquisition, R.M.; Methodology, K.X. and R.M.; Project administration, R.M.; Software, K.X., M.S. and D.W.; Supervision, R.M.; Validation, D.W.; Writing—original draft, K.X.; Writing—review & editing, K.X., R.M., D.W. and M.S.

**Funding:** This research was funded by State Key Program of National Natural Science of China (No. 41431176), National Natural Science Foundation of China (No. 41701416, 41771366), the Provincial Natural Science Foundation of Jiangsu of China (No. BK20181509), and the funding of NIGLAS (No. NIGLAS2017GH03).

**Acknowledgments:** The authors thank the colleagues from NIGLAS (Zhigang Cao, Yixuan Zhang, Minqi Hu, Tianci Qi, Junfeng Xiong, Qiao Chu, Jinge Ma, and Pengfei Zhan) for their help with field measurements and data collections. Acknowledgement for the data support from "Lake-Watershed Science Data Center, National Earth System Science Data Sharing Infrastructure, National Science & Technology Infrastructure of China. (http://lake.geodata.cn)".

**Conflicts of Interest:** The authors declare no conflict of interest.

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
