# Peer review of "Optical Classification of the Remote Sensing Reflectance and Its Application in Deriving the Specific Phytoplankton Absorption in Optically Complex Lakes"

_remotesensing, doi:10.3390/rs11020184_

Round 1

Reviewer 1 Report

Remote Sensing

Optical classification of the remote sensing… by Xue et al.

Aimed at:

1) identify the optical water types of the lakes in the LYHR Basin using field-measured and OLCI-derived Rrs(λ) data;

2) characterize the bio-optical properties and IOP variations 
in each OWT;

3) develop class-specific models to improve the estimation of the Chla content and 
Chla-specific phytoplankton absorption at 443 nm (aph*(443)). 

General comments

The authors wrote a good introduction, and stated “Our main interests were to characterize the optical variations in the 
lakes in the LYHR Basin and to assess the performance of class-specific bio-optical inversion models”, however, the authors need to justify in the introduction why the use of OWT is better than the other approaches. In fact, the authors need to conjecture about the problem and propose a hypothesis.

Specific comments

1) The authors used the rho value of 0.0028 (Mobley), but they don’t show that this method is the best for the dataset. Which is not acceptable. For example: Bernardo et al. (2018) showed that using the method from Lee et al. (2010) the error in estimate the Rrs was lower compared with others methods. There are others examples of published papers showing these experiments. 

Bernardo, N.; et al. Glint Removal Assessment to Estimate the Remote Sensing Reflectance in Inland Waters with Widely Differing Optical Properties. Remote Sens. 201810, 1655.

Lee, Z.; et al. Removal of surface-reflected light for the measurements of remote-sensing reflectance from an above-surface platform. Opt. Exp. 2010, 18, 26313–26324.

2) The authors used the k-means to classify the OWT, however, how about the other methods? Such as SVM, etc…? 

3) The authors wrote, “The 6SV model was proven to be more efficient than other 
atmospheric correction methods in turbid inland waters”. The authors need to show the results of the optical closure comparing the most used methods (e.g. Polymer, etc…).

4) How the authors transformed the OLCI surface reflectance into OLCI Rrs? Why the authors transformed the OLCI Rrs into log?

5) “In addition, the current optical classification tool built in SNAP software did not perform well due to the failure of atmospheric correction in the study region [44,45].” The authors tried to use the atmospheric corrected image and then use the SNAP tool? Explain.

6) The authors need to explain in detail how they clustered the data. Have you used endmembers? 

7) “The data at 400, 412, and 748 nm were not used due to the questionable accuracy of the atmospheric correction in inland waters”. My guess is because your results were bad at these wavelengths. This is why the authors need to show the comparison between the atmospheric correction methods.

8) The authors validated the specific-aphy estimation?

9) Table 2 – The scientific community has been working with OWT specially this year, because the development of bio-optical models for highly productive inland waters still a challenge. Was a surprise to see from Table 2 that the use of class did not improve the estimation. The errors were close to that using all dataset. This shows that for remote sensing of inland waters the development of bio-optical models for operational monitoring will be a challenge for long time.

10) Figure 6 is ok, because of that, the authors need to be clearer about the classification method used – please provide details to the readers.

Reviewer 2 Report

The paper meets the scope of the journal. It seems original and present a new popular topic related to inland water classification. Although the method is simple, using k-means, they had used S3A-OLCI dataset, which are the novelty of this work. Four types of water were found – one type with high effect of SPM; the other with high effect of phytoplankton, and two others represented mixed water types.  

When I checked your dataset, I raised some doubts: you sampled 535 locations, however, when we observe your Figure 1 we saw a lot of redundancies, it means, a lot of samples are overlapped (in the same location). Do the authors evaluate the spectral curves and remove the redundancies? I mean, some sampling spots can present the same Rrs spectra for different time acquisitions.

Line 13, p.1: What do you mean with “Labeled with the S3A-OLCI” in

Line 106-108:What is the definition of NAP/detritus? Are you certain that use a*d over SPM retrieved specific ad? Why did not use a*d/ SPIM?

Line 115: How did you converted Rrs into normalized Rrs? If you used a math, you can show it, because when you mention that you normalized Rrs by its integrated value in the spectrum range, what is the spectrum range? 350-1050?

Line 141: Dt2 = 11.2 based in statistics of all the OLCI images. What are these statistics?

Line 151: p(i) is the frequency or probability?

Line 161: I supposed that Chl-a indexes were changed by OLCI data, right? However, the 667 nm band was not used for OWT classification. What is the effect when the authors did not include 667 nm in OWT classification and use the same band for Chl-a retrieving?

Line 175: I think it is missing the RMSE definition (considering that you specified MAPE).

Line 224: I understand what you describe, however, this is confusing because ag contributes to 0.25 when compared to a­ph, for instance.

Table 2: Low differences between NR-2B and MER-3B for all OWT.

Line 288-290: It is missing something in the sentence.

Minor comments:

Section 3.2: It is important describe optical features from each OWT, however, take careful with a lot of details and small numbers used to describe absorption coefficients. Please, review this section to make it straight to the point.

Line 257: Type 3 had similar RMSE values compared to the overall dataset in the three algorithms, while type 4 had obviously larger RMSEs compared to the overall dataset. Maybe you can change to “Type 3 had similar RMSE values compared to the three algorithms, while type 4 had obviously larger RMSEs compared to the overall dataset”.

Figure 7. Datasets were available from Moore? Where did you got the data? Please, specified it.

Round 2

Reviewer 1 Report

For future works, I would like to suggest to the authors to do a sensitivity analysis to find out the main sources of errors in your results.